# Ionic Liquid-Mediated Interfacial Polymerization for Fabrication of Reverse Osmosis Membranes

**DOI:** 10.3390/membranes12111081

**Published:** 2022-10-31

**Authors:** Nisha Verma, Lexin Chen, Qinyi Fu, Skyler Wu, Benjamin S. Hsiao

**Affiliations:** Department of Chemistry, Stony Brook University, Stony Brook, New York, NY 11794, USA

**Keywords:** reverse osmosis membranes, interfacial polymerization, thin film composite, polyamide, ionic liquid, grazing incidence wide-angle X-rays scattering

## Abstract

This study revealed the effects of incorporating ionic liquid (IL) molecules: 1-ethyl, 1-butyl, and 1-octyl-3-methyl-imidazolium chlorides with different alkyl chain lengths, in interfacial polymerization (IP) on the structure and property (i.e., permeate-flux and salt rejection ratio) relationships of resulting RO membranes. The IL additive was added in the aqueous meta-phenylene diamine (MPD; 0.1% *w*/*v*) phase, which was subsequently reacted with trimesoyl chloride (TMC; 0.004% *w*/*v*) in the hexane phase to produce polyamide (PA) barrier layer. The structure of resulting free-standing PA thin films was characterized by grazing incidence wide-angle X-rays scattering (GIWAXS), which results were correlated with the performance of thin-film composite RO membranes having PA barrier layers prepared under the same IP conditions. Additionally, the membrane surface properties were characterized by zeta potential and water contact angle measurements. It was found that the membrane prepared by the longer chain IL molecule generally showed lower salt rejection ratio and higher permeation flux, possibly due to the inclusion of IL molecules in the PA scaffold. This hypothesis was supported by the GIWAXS results, where a self-assembled surfactant-like structure formed by IL with the longest aliphatic chain length was detected.

## 1. Introduction

Reverse osmosis (RO) desalination is an effective way to deal with the global scarcity of freshwater, especially in the coastal regions. This technique has been widely adopted because of its energy efficient performance and ability to provide potable water for diverse households, industrial and agricultural applications [1,2,3,4,5]. In the fabrication of RO membranes, varying membrane materials and formats have been demonstrated, including nanoporous cellulose acetate membranes and thin-film composite (TFC) membranes containing a polyamide (PA) barrier layer. Typical, the PA-TFC membranes can be operated over a wider pH range and at lower pressures than those used for cellulose acetate membranes [6,7,8].

The most essential component of the TFC composite membrane is the ultrathin PA barrier layer (with thickness in the range of several hundred nanometers), which is primarily responsible for the filtration performance. This PA layer is typically synthesized by interfacial polymerization involving the reaction between an aromatic diamine (e.g., m-phenylene diamine (MPD)) in the aqueous phase, and an acyl chloride (e.g., trimesoyl chloride (TMC)) in the immiscible organic phase. The structure of the cross-linked PA layer is highly sensitive to the polymerization conditions. Minor changes in the monomer types, concentration and volume, additives (e.g., surfactants), and other processing conditions can significantly affect the filtration performances of the resulting membranes [9,10,11,12].

Ionic liquids (ILs)-based surfactants have been used in preparation of filtration membranes as processing aides for varying water purification applications, such as wastewater treatment and potable water production [13]. Both anionic and cationic surfactants have been used in fabrication of RO membranes with the goal of increasing the permeation flux while maintaining high rejection ratio [14,15]. For example, Barabski-Karakby et al., modified the low-pressure RO membranes by grafting the material with poly(glycidyl methacrylate) using redox-initiated radical polymerization with a non-ionic surfactant (Triton X-100) to enhance the uniform monomer-coating [16]. They found that the inclusion of Triton X-100 could enhance the adherence of the grafted polymer to the membrane materials and increase the membrane surface polarization, which in turn reduced the monomer consumption and achieved a high reject ratio against sodium and chloride ions as well as boric acid [17]. Raval et al., investigated the chemical structural changes in RO membranes treated with different surfactants (anionic, cationic and non-ionic) and reported that the inclusion of different surfactants could result in an enhanced filtration performance [18]. Several other groups demonstrated that the inclusion of surfactants could also reduce the membrane fouling tendency. For example, Chen et al., reported the fouling reduction of ultrafiltration (UF) membranes for protein purification by incorporating a combination of surfactants during membrane fabrication [19]. In a similar study, Yamagiwa et al., confirmed the use of a series of non-ionic surfactants could prepare membranes with low fouling characteristics [20]. Wilbert et al., adopted this concept and used a homologous series of polyethylene-oxide surfactants to modify RO membranes, which also exhibited fouling resistance [21]. Finally, several studies further showed that the use of surfactants could qualitatively change the membrane surface property, such as roughness and wettability [22,23]. Unfortunately, the in-depth knowledge on the structure and property relationship by using the above surfactant-mediated interfacial polymerization approaches is still limited.

Conventional techniques to characterize the structure and functionality of PA barrier layers in RO membranes include Fourier transform infrared (FTIR) spectroscopy, scanning/transmission electron microscopy (SEM/TEM), atomic force microscopy (AFM), and X-ray photoelectron spectroscopy (XPS). However, considering that the barrier layers are made of highly cross-linked aromatic molecules and contain very small nanopore sizes (less than 1 nm), it is challenging to truly understand the porous structure at the molecular level with these techniques. Recently, some advances using synchrotron X-ray scattering techniques have been made in characterizing the ultrathin barrier layer structure [24]. For example, our group has demonstrated the use of grazing incidence wide-angle X-ray scattering (GIWAXS) technique to reveal the packing of molecular aromatic moiety in the polymer backbone, where two packing motifs (i.e., parallel and perpendicular) with a preferential surface-induced orientation were observed, which otherwise would not have been possible to reveal using other X-ray scattering techniques [25]. Foglia et al., used the X-ray and neutron reflectivity techniques to investigate the thickness of PA film exposed to different levels of H_2_O and D_2_O relative humidity (RH) and reported swelling and water uptake behavior of these thin films [26]. Sunday et al., further used soft X-rays to quantify the concentrations of different functional groups, namely, amide carbonyl and carboxylic acid, in PA films [27]. With small-angle X-ray scattering (SAXS) technique, Singh et al., reported that the polymer chains in the PA barrier layer facilitated the formation of nanostructures, which were interconnected and formed nanoclusters [28]. Pipich et al., also demonstrated the use of small-angle neutron scattering (SANS) measurements to characterize PA films and revealed the surface morphology and its relationship with interconnected pore distributions [29].

In this study, we have investigated the fabrication of PA barrier layers in RO membranes synthesized by using the ionic liquid (IL)-mediated interfacial polymerization approach. The approach involves the dispersion of cationic IL molecules containing the same imidazolium hydrophilic head but different alkyl chain lengths, in the aqueous solution of MPD. The effect of IL molecules with different alkyl chain length (the longest alkyl IL molecule behaves as a surfactant) on the structure of resulting PA layers was studied by the GIWAXS technique. The objective of this study is to understand the effects of incorporating imidazolium-based IL molecules, from inert shuffling agent to surfactant, in interfacial polymerization on the structure and property (i.e., permeate-flux and salt rejection ratio) relationships of resulting RO membranes. In specific, we aim to understand if the imidazolium-based molecules can be trapped and/or self-assembled in the PA scaffold, leading to changes in the barrier layer structure, filtration performance and membrane surface properties (e.g., hydrophilicity and charge). It should be noted that some previous studies have dealt with the similar thin film composite membrane system. However, no structural studies on the polyamide barrier layer at the molecular or nanoscale level by using GIWAXS have ever been carried out. Additionally, no information has been reported for the solute transport by relating it with the aromatic motif orientation in these barrier layers, which is the focus of the present study.

## 2. Ionic Liquid Selection

Three ionic liquid molecules were chosen for this study: (a) EMIC (1-ethyl-3-methyl-imidazolium chloride, molecular weight = 146.62 g/mol) with 2 carbon atoms, (b) BMIC (1-butyl-3-methyl-imidazolium chloride, moleular weight = 174.67 g/mol) with 4 carbon atoms, and (c) OMIC (1-octyl-3-methyl-imidazolium chloride, molecular weight = 230.78 g/mol) with 8 carbon atoms. Our rationale for explore the IL-mediated interfacial polymerization study is that OMIC can behave as a surfactant molecule, which can be aligned at the water/solvent interface because of the amphiphilic nature. In other words, the hydrophobic tails of the OMIC molecules should be immersed in the organic phase and their hydrophilic heads are in the aqueous phase, where the self-assembly process at the interface may facilitate the MPD diffusion and create an oriented structure during the formation of the PA barrier layer. In contrast, EMIC and BMIC should not exhibit the surfactant behavior and behave as a chemical agent. In this case, no self-assembled structures of BMIC and EMIC can be formed at the interface, but these molecules can attract MPD and shuffle them across the interface due to the higher solubilities of BMIC and EMIC in hexane [30] (MPD is nearly insoluble in hexane) and the favorable attractive interaction (π-π stacking) between the aromatic groups between MPD (benzene ring) and BMIC/EMIC (imidazolium ring).

In a study of [C_4_mim][PF_6_] surfactants, the liquid-like (LL) model was applied to determine the molecular volume (V_m_) as a function of mass density, Avogadro number, and molecular weight [31]. This approach was also used to determine V_m_ of the chosen surfactants. Figure 1a,b show the structure of the surfactants in this study, and the plots for V_m_ against the number of carbon atoms in the alkyl chain for both [C_4_mim][PF_6_] [32] and EMIC/BMIC/OMIC surfactants, respectively. A linear dependency of V_m_ change with the number of carbon atom was observed in the selected surfactants. The slopes from both sets of the surfactants were approximately the same, and they could be used to determine the effective volume for each -CH_2_- moiety. These results indicate that a uniform alkyl slab/liquid-like structure is also present in the EMIC/BMIC/OMIC surfactants on the nanoscale.

## 3. Experimental

### 3.1. Materials

1,3,5-Benzenetricarbonyl trichloride (TMC, 98.0+%, TCI America), m-phenylenediamine (MPD, 99+%, ACROS Organics), 1-octyl-3-methyl-imidazolium chloride (OMIC, 97%), and 1-butyl-3 methyl-imidazolium chloride (BMIC, 96%) were purchased from Alfa Aesar. 1-Ethyl-3-methyl imidazolium chloride (EMIC, 97%) and ethanol (99.5%) were purchased from Acros Organics. All chemicals were used as received without further purification unless otherwise noted. The polysulfone (Psf) ultrafiltration (UF) flat membrane sheets (US020) with a molecular weight cut-off of 20,000 g/mol was used as an RO membrane support was purchased from RisingSun Membrane Technology Co., Ltd. (Beijing, China).

### 3.2. IL-Mediated Interfacial Polymerization (IP) for RO Membrane Fabrication

The as-received US020 substrate (cut in a 11” × 9” sheet) was sequentially immersed in ethanol for 5 min and then in 0.1% (*w*/*v*) MPD aqueous solution containing the IL molecules (i.e., EMIC, BMIC, OMIC (1% *w*/*v*) for 2 min. The MPD-soaked membrane was carefully taped onto a clean glass plate. Excess MPD solution was removed using a glass rod. Subsequently, the MPD-soaked supported membrane was immersed in a 0.04% (*w*/*v*) TMC solution for 2 min. The tape was carefully peeled off and the interfacially polymerized membrane was placed in an oven at 70 °C for 10 min. A schematic diagram for the preparation of the RO membrane is depicted in Figure 2. The resulting membranes were evaluated for the RO filtration performance using a crossflow apparatus to be described below. Zeta potential and contact angle measurements were also carried out to characterize these membranes.

### 3.3. RO Filtration Evaluation

A bench-scale crossflow device was used to simultaneously evaluate the RO performance (water permeation flux and salt rejection ratio) of six prepared membranes (Figure 3). The effective membrane area for the measurement was 36 cm^2^. All membranes were first subjected to simulated seawater prepared using a 3.5% (*w*/*v*) NaCl solution in DI water for 30 min at 800 psi, regardless of the IL molecules used, and then operated for another 5 h at 800 psi to measure water permeability and salt rejection ratio. In the filtration measurement, the crossflow unit was first stabilized with deionized water at a hydraulic pressure of 1 bar for 1 h. The volume of the filtrate was measured to determine the water permeation flux (L m^−2^ h^−1^; LMH), where the salt rejection, R was determined as (1 − C_p_/C_f_), where C_p_ is the salt concentration of the permeate and C_f_ is the salt concentration of the feed solution. The permeation flux was measured for 5 h to evaluate the stability of the membrane. The reported results of the permeation flux and rejection ratio were the averaged values of three independent measurements.

### 3.4. Preparation of Free-Standing Thin Films

To probe the structure of the PA barrier layer in the RO membranes, free-standing thin films were also prepared using the same IL-mediated IP methods, as illustrated in Figure 4. Briefly, 5 mL of 0.1% (*w*/*v*) MPD solution thoroughly mixed with 1% (*w*/*v*) IL was poured in a 50 mL beaker containing a clean 3 cm × 2 cm silicon wafer at the bottom of the beaker. Approximately 2 mL of pure hexane was added dropwise as a buffer layer to ensure a smooth formation of thin film at the interface. Next, 1 mL of 0.004% (*w*/*v*) of TMC solution was carefully added dropwise onto the buffer layer. After 2 h, excess solution was drained using a pumping syringe, gradually letting the film to settle over the silicon wafer. The film was then treated with 5 mL of ultrapure water for 6 h. Excess liquid was drained again, and the silicon wafer was left to dry overnight for further use.

### 3.5. GIWAXS Characterization

Two-dimensional (2D) GIWAXS measurements of the PA thin films prepared by the IL-mediated IP method were carried out in the Complex Materials Scattering (CMS) beamline (11-BM) at the National Synchrotron Light Source II (NSLS-II), Brookhaven National Laboratory. The X-ray patterns were collected by Pilatus (Dectris) detector and the data acquisition time for each GIWAXS pattern was from 5 to 30 s. A slit limited beam size of 0.05 by 0.2 mm^2^ was used to illuminate the sample using a 0.0918 nm X-ray wavelength and at a grazing angle of 0.12°, close to the critical angle of the underlying silicon wafer where the signal is enhanced. These are operated in the sample vacuum chamber as the sample, thus minimizing the background scattering. The sample to detector distance was 22.9 cm. Details of the data analysis procedures have been previously described using two-Lorentzian profiles to extract relevant structured parameters regarding the aromatic motif arrangements in the PA chains [25].

### 3.6. Contact Angle Measurement

The contact angle of a water droplet on the IL-mediated interfacially polymerized membrane surface was measured using an optical contact angle analyzer (CAM200, KSV instrument, LID, Helsinki, Finland). Approximately 5 μL-water droplet was gently placed on the surface of the membrane, and digital images were collected using the CAM software. The images were recorded after a fixed time of 5 s. The water contact angle was determined using the curve fitting method and was reported as an average value at 3 different locations. The experiments were repeated three times to check the reproducibility.

### 3.7. Zeta Potential Measurement

A zeta potential analyzer (Anton Paar, SurPASS 3, Graz, Austria) was used to study the surface charge of the membrane fabricated using different ILs at a fixed concentration. In this measurement, the membranes with dimensions of 20 mm × 10 mm was glued onto an adjustable gap cell, maintaining a gap distance of approximately 100 µm. The streaming potential measurements were carried out over the pH range of 3.5–9.0. The reported value was also the averaged results from 3 independent runs.

## 4. Results and Discussion

### 4.1. Performance Evaluation of IL-Mediated Interfacially Polymerized RO Membranes

Figure 5 describes the NaCl rejection ratio and permeation flux of IP membranes as a function of EMIC, BMIC, OMIC concentrations used during the preparation procedures, along with those of commercial membranes. The results clearly indicated that the inclusion of all IL concentrations in the aqueous MPD monomer yielded a higher permeation flux but lower rejection ratio. As shown in Figure 5a–c, the IL-mediated IP RO membranes increased the permeation flux by around 50% (over 22 L m^2^-h^−1^) but decreased the rejection ratio by around 10% with 5% of IL in interfacial polymerization. Such a trade-off between the rejection ratio and permeation flux is a common characteristic in RO membranes.

Figure 5a–c also show the effects of different ILs and their concentrations (1–5 *w*/*v*%) on the membrane performance. EMIC being the smallest molecule showed the highest salt rejection and lowest water flux, while OMIC being the longest molecule showed the lowest salt rejection and highest water flux with an average flux of ~34.50 L m^2^-h^−1^. On the contrary, the BMIC-modified membrane showed a mean permeation flux of ~25.57 L m^2^-h^−1^ and EMIC showed a mean flux of ~24.00 L m^2^-h^−1^. However, the rejection ratio of EMIC (~98.90%) was found to be higher than those of BMIC (~95%) and OMIC (~92%). The different behavior can be attributed to the different content of the IL molecule trapped in the PA scaffold, depending on its chain length, leading to different rejection ratio against Na and Cl ions. It was found that the increase in the alkyl chain length resulted in the increasing water flux but decreasing salt rejection ratio.

Regarding the effect of chain length of the IL molecule, the membranes fabricated with the longest chain length OMIC exhibited the highest permeation flux but lowest rejection ratio (Figure 5). There are several possibilities that can explain this observation. Since OMIC molecules are typical surfactant molecules, they can preferentially migrate to the water/organic solvent interface during IP and lower the interfacial surface tension. This process would facilitate the transport of MPD molecules to the organic phase and initiate the reaction with TMC. It is conceivable that as the surfactant concentration increases, the tendency to form micellar structure in the aqueous phase and self-assembled layered structure at the interface would increase, whereby the latter could also hinder the transport of MPD to the organic phase and resulting in some entrapment of the OMIC layered assembly in the PA scaffold. In this scenario, the existence of hydrophobic tail aggregates are likely to enlarge the free volume space within the PA matrix that would increase the permeation flux and decrease the rejection ratio. These hypotheses are consistent with the observed filtration results in Figure 5c, and our earlier study of OMIC-based RO membranes [33]. We further speculate that once these OMIC molecules are incorporated in the PA scaffold, they may be difficult to be removed by conventional washing methods. This is because the sizes of the OMIC assemblies may be too large to be extracted away and the assembly may be anchored by strong attractive interaction forces between the aromatic groups between PA and OMIC. To further understand the relationship between the filtration performance and the structure of the PA barrier, GIWAXS results of IL-mediated interfacial polymerized PA films will be discussed in the next section.

In a different perspective, EMIC and BMIC with short aliphatic chain lengths cannot exhibit surfactant behavior, but they can serve as chemical carriers to shuffle MPD from the aqueous phase to the organic phase and facilitate the polymerization process between MPD and TMC. This is because the solubilities of EMIC and BMIC in hexane are higher than MPD in hexane due to their aliphatic tails. As a result, the presence of EMIC and BMIC molecules can increase the diffusion rate of MPD across the interface into the organic (hexane) phase. This hypothesis is based on a previous study which reported that the ammonium ionic compound (having a similar structure as EMIC and BMIC) acted as a carrier of the reactant species from the aqueous phase to the organic phase and facilitated the polymerization reaction [34]. In Figure 5a,b, it was seen that the increase in EMIC or BMIC concentration all resulted in membranes with higher permeation rate and lower rejection ratio, which are certainly consistent with the concept of the shuffling agent.

In Figure 5a–c, it was seen that the membrane prepared by the longer chain IL molecule generally showed lower salt rejection ratio and higher permeation flux. One possible reason for this observation is due to the higher solubility of OMIC (because of its longer alkyl chain) in hexane than those of BMIC and EMIC. The relatively higher solubility would allow more OMIC molecules to migrate into the hexane phase, thereby causing a higher degree of OMIC entrapment in the PA scaffold and preventing the tightening of the network formation. Based on the flux data, the average water permeances for EMIC, BMIC, and OMIC-based membranes are calculated as 0.447, 0.489, and 0.642 L m^−2^ h^−1^ bar^−1^, respectively [35]. Similarly, the corresponding salt permeability coefficients are calculated as 0.787, 1.22, and 2.732 L m^−2^ h^−1^, respectively [36]. To this end, Figure 5d presents the comparative data of the membranes synthesized in this study using IL and the commercially available RO membranes. Although the commercial membranes exhibit relatively higher permeation flux, the IL-mediated membranes fabricated in this study show larger salt rejection ratio.

### 4.2. GIWAXS Study of IL-Mediated IP PA Films

Figure 6 illustrates the GIWAXS results, including the as-measured scattering images, circularly averaged scattering profiles and the Lorentzian fits (black line), and the scattered intensity versus χ plots of free-standing PA thin films synthesized with the aid of EMIC, BMIC and OMIC. These results revealed some new insights into the roles of these IL molecules during interfacial polymerization, and how they might affect the cross-linked network of the barrier layer. The as-measured 2D GIWAXS patterns for PA thin films synthesized using EMIC, BMIC and OMIC are shown in Figure 6A–C, respectively, and without the surfactant molecules (the layer thickness has been reported to be around 8–24 nm using neutron and X-ray reflectivity measurements [26]). These images can be interpreted as follows.

The 2D GIWAXS scattering patterns in Figure 6A–C show uniform scattering rings with equal intensity at all azimuthal angles, irrespective of the ILs incorporated. The scattering feature is broad in width and can be perceived as of an amorphous phase with a short-range order. The q_r_ and q_z_ labels represent the parallel and normal scattering wave vectors for the film surface, respectively, on the *y*-axis and *x*-axis. These scattering features are evidently radially diffused and are of rather isotropic characteristics. The scattered intensity profiles extracted at three selected azimuthal angles (χ = 10, 40, and 80°) for each film are shown in Figure 6D–F, where these linear profiles were fitted as the sum of two independent Lorentzian curves along with a sloping background. The rationale and the fitting procedure to obtain these structural parameters have been described in our previous study [25]. Briefly, the PA chains in the barrier layer are amorphous, whereby the observed GIWAXS patterns (after the background subtraction) indicted the molecular packing of polymer chains. In the previous study, two distinct peaks were identified in the GIWAXS profiles, which were attributed to two different packing motifs of the aromatic moiety: parallel stacking and perpendicular stacking (or T-shape stacking), both could exhibit some preferred orientation [25], in the PA network. Moreover, molecular spacing was closer packed in orientations along the surface normal direction than in orientations along the surface in-plane direction.

It is interesting to see in the present spectra (Figure 6D–F), one fitted peak (*q*_1_) was centered around 1.62 Å^−1^, while the other peak (*q*_2_) was centered around 1.83 Å^−1^, irrespective of the IL molecules used. The d-spacings corresponding to these two *q*-values (using Bragg’s law) were 3.5 and 3.8 Å, respectively. Both values indicate the presence of parallel π-π stacking of aromatic moieties in the PA scaffold, perhaps in two different populations [25]. Unfortunately, the presence of the perpendicular packing motif (or T-shape motif), typically observed at *q* = 1.22 Å^−1^ (or d = 5 Å) is not observed. The maximum scattered intensities of the two Lorentzian fits at different χ values are found to be approximately constant (Figure 6G–I), implying a random distribution of parallel stacking of aromatic moieties. These features were seen in all PA films, indicating that the presence of IL molecules promotes the random distribution of parallel motif and hinders the formation of perpendicular motif. Therefore, it is unlikely for a water molecule (diameter about 2.75 Å) to transport through two parallelly stacked aromatic rings with a separation distance of 3.5–3.8 Å; therefore, there must be the presence of larger molecular channels formed in the loosely cross-linked PA network.

The RO membranes based on EMIC and BMIC (short alkyl chain lengths) indicate a high rejection ratio, as discussed earlier. However, such an effect diminishes in the RO membranes synthesized using OMIC (with 8-carbon atoms). As seen in Figure 6C, an intense low-*q* scattering peak appears at around 0.2 Å^−1^ (d ~31.4 Å). This peak indicates that the OMIC layered assemblies were trapped within the PA scaffold. Although the initial concentration of OMIC in the aqueous MPD phase was below the critical micelle concentration (CMC ~20%), the formation of PA chains increases the concentration trapped OMIC molecules, resulting in a possible micellar formation. In this case, the low-*q* scattering peak (0.2 Å^−1^) reflects the dimension of the self-assembled OMIC molecules, formed by the aggregation of hydrophobic tails and parallel stacking of aromatic heads (i.e., the imidazolium group) with the surrounding benzene groups in PA chains. This low-*q* scattering peak is often referred as the “pre-peak” in imidazolium-based ionic liquids, which has been explained via the mesoscopic organization of molecules leading to nanoscale segregation [37]. The presence of the pre-peak in OMIC also confirms that the layered structure is present in the PA matrix during the IP process and such assembly cannot be washed away. In contrast, no low-*q* scattering peak can be found in PA thin films fabricated using EMIC and BMIC. Therefore, if EMIC and BMIC molecules are also trapped within the PA matrix, these two molecules lack the self-assembly capability as they are not surfactant molecules. The incorporation of EMIC/BMIC/OMIC will all decrease the cross-linking density of the PA scaffold, thus promoting the water flux but reducing the rejection ratio. The self-assembly behavior of OMIC molecules can further decrease the cross-linking density [38] and cause a much looser network work structure. It is conceivable that the OMIC’s capacity to self-assemble aromatic heads together during the IP process leads to OMIC being stacked in a parallel fashion. The occurrence of a parallel shaped motif thus may contain benzene-benzene stacking, benzene-imidazolium stacking and imidazolium-imidazolium stacking. As a result, T-shaped motifs in the PA matrix may be overshadowed by the high presence of parallel shaped motifs and thus disappear. As a result, the membranes containing OMIC assemblies have no preferred orientation, which limits the degree of cross-linking. As observed in the OMIC-mediated membrane filtration data, a low cross-linking PA matrix favors the formation of water channels with high flux but low selectivity. Table 1 sumarrizes the GIWAXS characterization data and the membrane performance indicators.

### 4.3. Water Contact Angle Measurements of IL-Mediated IP RO Membranes

Figure 7 shows the water-contact angle results from the RO membranes prepared by using different IL molecules. The data showed an increase in hydrophilicity of the membranes with increasing hydrophobic chain lengths of ILs. This is in contrast with the hydrophobicity of the pure ILs, which increases in the order EMIC < BMIC < OMIC because of the increase in the number of saturated (-CH_2_-) bonds. However, it was interesting to observe that when the IL molecules were included in the aqueous MPD solution during interfacial polymerization, the trend reversed. The reason that membranes fabricated using OMIC exhibited more hydrophilic characteristics than those with BMIC and EMIC can be as follows. The number of the amide bonds in PA formed from interfacial polymerization is mainly responsible for the hydrophilicity of the membrane. In addition, the free carboxylic group or unreacted acyl chloride in TMC in the barrier layer can also contribute to the hydrophilicity of the membranes. It is conceivable that the membranes prepared by OMIC contains trapped OMIC assemblies, creating a loos network structure with more unreacted carboxylic groups than those prepared with EMIC and BMIC molecules. In other words, the smaller IL molecules with shorter alkyl chains can create the PA scaffold with a tighter network structure and fewer unreacted chain ends (carboxylic groups) and thus more pronounced hydrophobic behavior.

### 4.4. Zeta Potentials of IL-Mediated IP RO Membranes

Figure 8 shows the zeta potential values of the RO membranes fabricated using ILs at different pH values. The tested membranes were prepared using 0.1% (*w*/*v*) MPD and 0.04% (*w*/*v*) TMC, while the IL concentration was maintained at a constant value of 5% *w*/*v*. It was seen that the changes in the isoeclectric points (around pH = 3.0) were insignificant because these IL molecules did not participate in the polymerization reaction. In addition, the zeta potentials remained negative in the pH range of 3–10 for all samples at a fixed IL concentration. However, it was interesting to observe that the zeta potential in the neutral pH range was approximately −10 V for OMIC, which decreased significantly to approxiamtely −35 V for both EMIC and BMIC. In contrast, for the pristine membrane fabricated without IL, its zeta potential curve was between those of OMIC and BMIC/EMIC. For the large chain length molecule, i.e., OMIC, the zeta potential values were positive at low pH values (pH < 4).

The difference between the surface charges of these membranes can be explained by the existence of unreacted monomers (e.g., MPD) and adsorbed IL molecules in the PA barrier layer. Both moieties are anionic and can affect the charge density on the membrane surface, despite not being part of the PA chains. In specific, the negatively charged carboxyl functional groups in the PA scaffold can be formed by hydrolysis of unreacted acyl chloride groups in TMC [39]. As the pH value increases, the acidic carboxyl groups can deprotonate, therefore decreasing the zeta potential. This explanation is consistent with the results obtained in Figure 8. Furthermore, since the zeta potential of the pristine membrane (without IL) was always higher than those of the membranes fabricated by EMIC and BMIC, this observation indicated that these anionic IL molecules were probably trapped within the PA scaffold. This is because the presence of EMIC and BMIC molecules in the PA scaffold could further decrease the zeta potential of the membrane, even though their existence cannot be detected by the GIWAXS method. In contrast, the presence of the OMIC assemblies was verified in the PA layer (Figure 6C). However, the formation of OMIC self-assembled structure did not reduce the membrane zeta potential when compared with that of the pristine membrane. Furthermore, the zeta potential decline of the membrane having OMIC became smaller with increasing pH values, when compared to the zeta potential decline in other membranes, because of the OMIC assembly being stable at higher pH values, resulting in a less charged surface.

## 5. Conclusions

The current study provides some new insights into the effects of adding ionic liquids in interfacial polymerization to prepare RO membranes, based on the combined structural and filtration performance studies. The filtration study indicated the incorporation of IL generally increases the permeate flux but decreases the salt rejection rate. OMIC being the longest chosen IL molecule with surfactant characteristics showed the lowest salt rejection (94.25%) and the highest water flux of 34.50 L m^2^-h^−1^, whereas EMIC, the shortest chosen IL molecule showed the lowest flux of ~24.00 L m^2^-h^−1^ and the highest rejection rate (~95%) compared to the BMIC and OMIC-mediated RO membranes. The differences between the alkyl chain lengths in IL molecules lead to different roles of ILs can play during interfacial polymerization, from molecular shuffling agent to a surfactant. After interfacial polymerization, some IL molecules appear to be trapped within the PA scaffolds, creating a loose cross-linked structure leading to an increase in permeation flux and a decrease in rejection ratio results. The evidence of the IL entrapment is evident by the existence of OMIC layered assembly in the PA scaffold, and the notably zeta potential decrease in membranes prepared with EMIC and OMIC. The entrapped IL molecules cannot be easily removed by conventional washing process, and they appear to be stable during RO filtration. This study has shown the GWAX analysis of the structure-performance relationship in the IL-mediated membranes to be useful in selecting an appropriate IL, in particular the effect of chain length on permeate flux and salt rejection ratio.

## Figures and Tables

**Figure 1 membranes-12-01081-f001:**
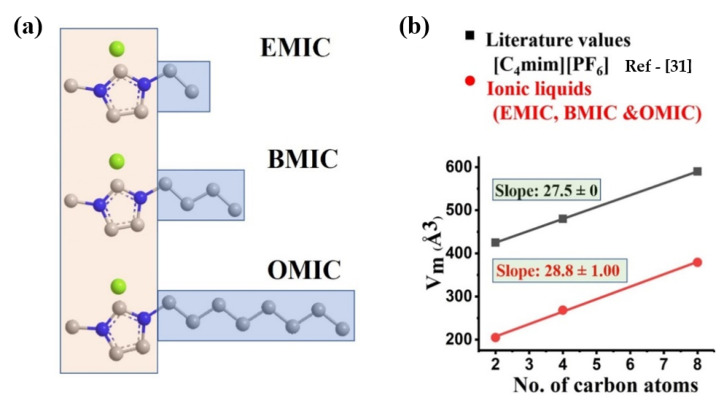
Schematics of (**a**) the surfactant molecule structures showing different lengths in increasing orders and (**b**) their molecular volumes as a function of carbon atoms.

**Figure 2 membranes-12-01081-f002:**
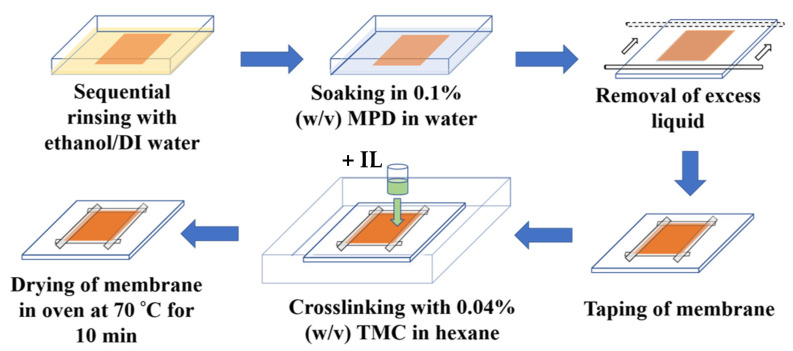
Preparation diagrams for the fabrication of RO membranes using the IL-mediated interfacial polymerization approach.

**Figure 3 membranes-12-01081-f003:**
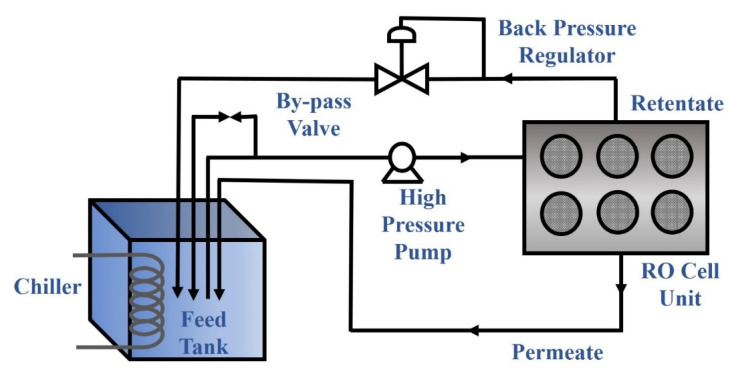
Schematics of the RO crossflow desalination setup to measure the filtration performance of the IL-mediated interfacially polymerized membranes.

**Figure 4 membranes-12-01081-f004:**
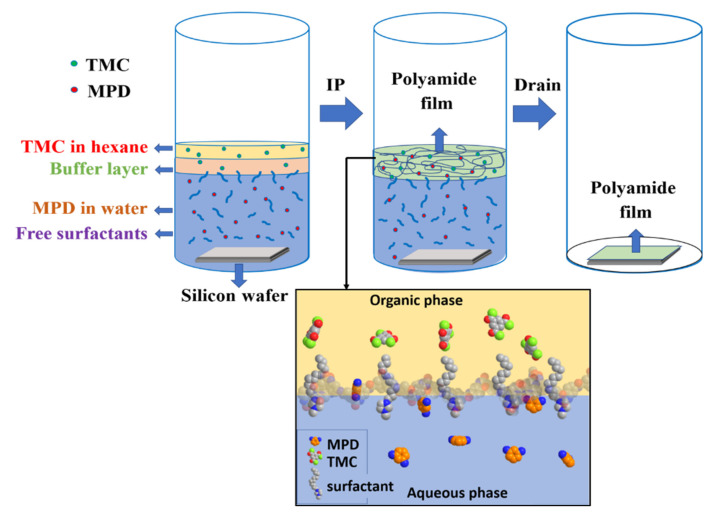
The sample preparation diagram for free-standing PA films using the IL-mediated IP method.

**Figure 5 membranes-12-01081-f005:**
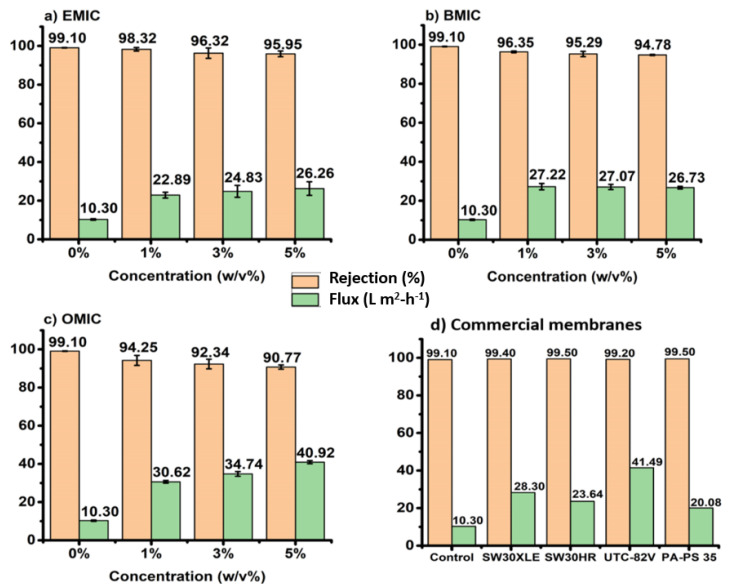
Dependence of NaCl rejection ratio and permeation flux as a function of (**a**) EMIC (**b**) BMIC (**c**) OMIC concentrations, and (**d**) commercial membranes. The samples prepared using 0% IL are used as control.

**Figure 6 membranes-12-01081-f006:**
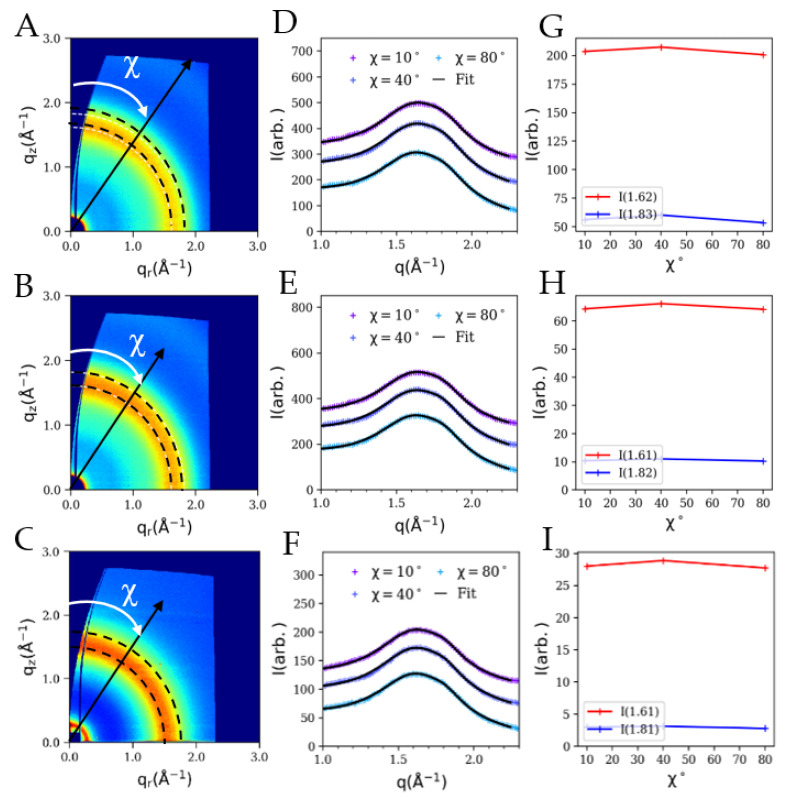
(**A**–**C**) 2D GIWAXS images, (**D**–**F**) circularly averaged scattering profiles (blue points) along with the fits using two Lorentzian functions (black line), and (**G**–**I**) the maximum scattered intensities of the two fitted peaks versus χ plots of the PA thin films synthesized with the aid of EMIC (5% *w*/*v*), BMIC (5% *w*/*v*) and OMIC (5% *w*/*v*), respectively.

**Figure 7 membranes-12-01081-f007:**
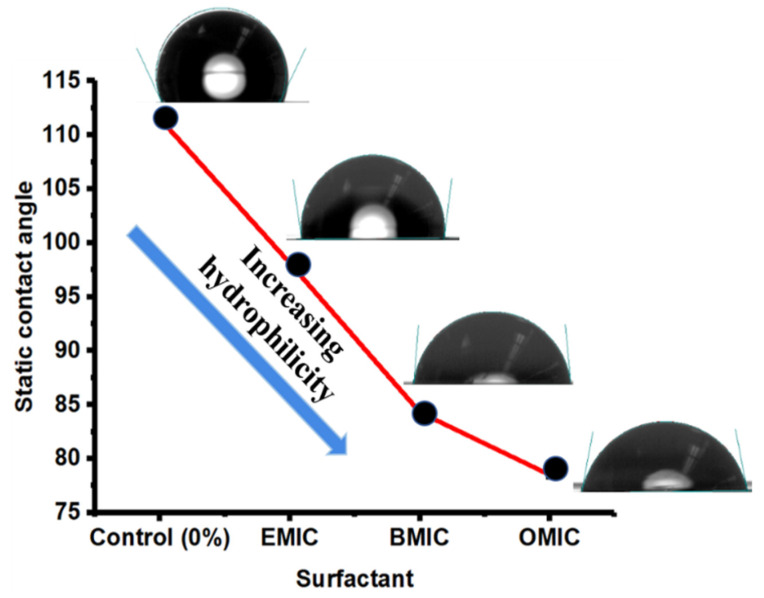
Water-contact angles of the IL-mediated IP RO membranes using different IL molecules.

**Figure 8 membranes-12-01081-f008:**
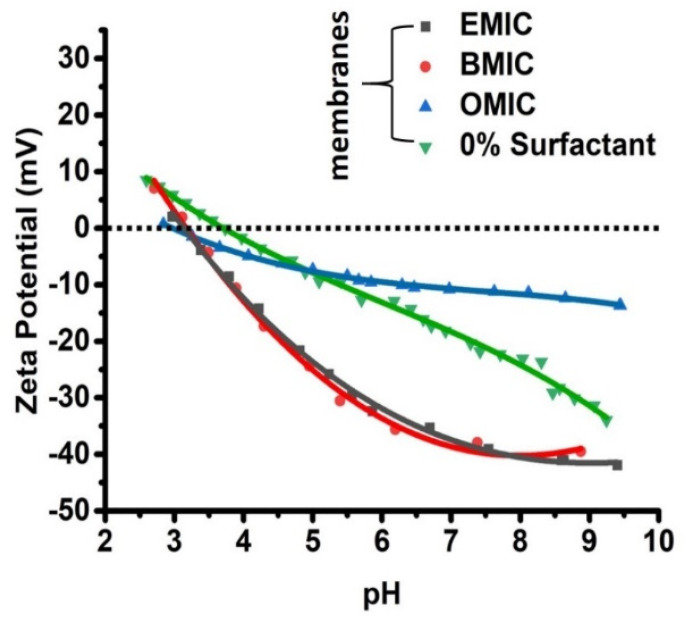
Zeta potentials representing the isoelectric points of the IL molecules.

**Table 1 membranes-12-01081-t001:** Peak positions and resulting d-spacings of free-standing films fabricated using different ILs (at different concentrations), and the corresponding measured rejection and flux.

IL (in Thin PA Film)	Alkyl Chain Length	*q*_1_ (Å^−1^)	*q*_2_ (Å^−1^)	d_1_ (Å)	d_2_ (Å)	R (%)	F (L m^2^-h^−1)^
EMIC ^1^	2	1.62	1.83	3.8	3.6	98.32	22.89
BMIC ^2^	4	1.61	1.82	3.9	3.4	95.29	27.07
OMIC ^3^	8	1.61	1.81	3.9	3.4	92.34	34.73

^1^ with EMIC; ^2^ BMIC; ^3^ OMIC, each 5% *w*/*v*.

## Data Availability

Not applicable.

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
