# Peer review of "Ionic Liquid-Mediated Interfacial Polymerization for Fabrication of Reverse Osmosis Membranes"

_membranes, 2022, doi:10.3390/membranes12111081_

Round 1

Reviewer 1 Report

The manuscript prepared by Verma et al. described a work using ionic liquid in the interfacial polymerization process to synthesize RO membrane. Three ILs with different chain lengths were selected. The authors demonstrated that the IL length was positively correlated with water permeability and negatively correlated with salt rejection. To explain this, GIWAXS technique was applied. The experimental design was good, the data presentation is sufficient to support its conclusion. Most importantly, the explanation and discussion regarding experimental data are beautiful. I would recommend acceptance of this manuscript after minor revision.

1.      Can the authors introduce more about the application of using ILs in assisting the synthesis of polyamide membrane in the section of Introduction? I feel sudden when I started reading after line 91.

2.      The membrane characterization needs to be revised. In general, to characterize RO membrane, the values of A (water permeance, L m-2 h-1 bar-1) and B (salt permeability coefficient, L m-2 h-1) need to be reported. A value can be calculated from the slope of the water fluxes as a function of the applied pressure (J. Membr. Sci.618 (2021), p. 118568), and B can be determined with the water flux and the intrinsic salt rejection rate (Environ. Sci. Technol. Lett.3 (2016), pp. 112-120). Can the authors provide the values of A and B for characterizing RO membranes?

Author Response

We thank the reviewer for recommending the manuscript for publication. The point-by-point response to the reviewer's minor comments is as follows:

  • Can the authors introduce more about the application of using ILs in assisting the synthesis of polyamide membrane in the section of Introduction? I feel sudden when I started reading after line 91.

We thank the reviewer for the comment and suggestion. Please note that ionic liquid is also termed a surfactant that carries an ionic group in the form of either a negative or positive head. We have now clarified this aspect in the manuscript when first referring to ionic liquids or surfactants.

"Ionic liquids (ILs)-based surfactants are commonly used in preparation of filtration membranes as processing aides for varying water purification applications..."

  • The membrane characterization needs to be revised. In general, to characterize RO membrane, the values of A (water permeance, L m-2 h-1 bar-1) and B (salt permeability coefficient, L m-2 h-1) need to be reported. A value can be calculated from the slope of the water fluxes as a function of the applied pressure ( Membr. Sci., 618(2021), p. 118568), and B can be determined with the water flux and the intrinsic salt rejection rate (Environ. Sci. Technol. Lett., 3 (2016), pp. 112-120). Can the authors provide the values of A and B for characterizing RO membranes?

We thank for the suggestion. We have now calculated the values of water permeance and salt permeability coefficients for different membranes as per the procedure referred by the reviewer. These values are now reported in the revised manuscript.

"Based on the data shown for flux, the average water permeances for EMIC, BMIC, and OMIC-based membranes are calculated as 0.447, 0.489, and 0.642 L m-2 h-1 bar-1, respectively []. Similarly, the corresponding salt permeability coefficients are calculated as 0.787, 1.22, and 2.732 L m-2 h-1, respectively []."

Reviewer 2 Report

Overall the article is well written, scientifically the quality is also acceptable. However following improvements can be made:

As the membranes under consideration are Thin Film Composite (TFC) and are the main focus of study, so key words may be modified accordingly.

Novelty of the work needs to be addressed, already several researchers done similar (as mentioned below) work a decade ago, How this work is different from those?

Yung, L., Ma, H., Wang, X., Yoon, K., Wang, R., Hsiao, B. S., & Chu, B. (2010). Fabrication of thin-film nanofibrous composite membranes by interfacial polymerization using ionic liquids as additives. Journal of Membrane Science365(1-2), 52-58.

Mariën, H., & Vankelecom, I. F. (2018). Optimization of the ionic liquid-based interfacial polymerization system for the preparation of high-performance, low-fouling RO membranes. Journal of Membrane Science556, 342-351.

Hartanto, Y., Corvilain, M., Mariën, H., Janssen, J., & Vankelecom, I. F. (2020). Interfacial polymerization of thin-film composite forward osmosis membranes using ionic liquids as organic reagent phase. Journal of Membrane Science601, 117869.

In regards to characterization of membranes, still Scanning Electron Microscopy (SEM) is relevant as it may give important information regarding surface morphology as well as inner structure of membranes, so SEM results at various magnification may be added. 

The performance of the membranes is only studied at a pressure of 1 bar while for Reverse Osmosis the pressure may go higher. So the studies at higher pressure may be reported.

Author Response

We thank the reviewer for recommending the manuscript for publication. The point-by-point response to the reviewer's minor comments is as follows.

Q1 As the membranes under consideration are Thin Film Composite (TFC) and are the main focus of study, so key words may be modified accordingly.

We thank the reviewer for the suggestion. We have now included the suggested "thin film composite" in the list of keywords.

Q2   Novelty of the work needs to be addressed, already several researchers done similar (as mentioned below) work a decade ago, How this work is different from those?

Yung, L., Ma, H., Wang, X., Yoon, K., Wang, R., Hsiao, B. S., & Chu, B. (2010). Fabrication of thin-film nanofibrous composite membranes by interfacial polymerization using ionic liquids as additives. Journal of Membrane Science365(1-2), 52-58.

Mariën, H., & Vankelecom, I. F. (2018). Optimization of the ionic liquid-based interfacial polymerization system for the preparation of high-performance, low-fouling RO membranes. Journal of Membrane Science556, 342-351.

Hartanto, Y., Corvilain, M., Mariën, H., Janssen, J., & Vankelecom, I. F. (2020). Interfacial polymerization of thin-film composite forward osmosis membranes using ionic liquids as organic reagent phase. Journal of Membrane Science601, 117869.

We have carefully gone through studies described in the suggested references, and have notes that these studies including from our research group deal with the similar thin film composite materials. However, none of them have studied the polyamide/barrier layer at molecular or nanometer level in the IL-based membranes using GIWAXS. Also, no information were provided for the solute transport by relating it with motif orientation, which we have now done in the present study. This is also the novelty of our study. We have now explicitly clarified this aspect in the manuscript.

"This may be noted that all previous studies deal with the similar thin film composite materials, however, without studying the polyamide/barrier layer at molecular or nanometer level in the IL-based membranes using GIWAXS. Also, no information have been reported for the solute transport by relating it with motif orientation, which is the focus of the present study."

 Q2 In regards to characterization of membranes, still Scanning Electron Microscopy (SEM) is relevant as it may give important information regarding surface morphology as well as inner structure of membranes, so SEM results at various magnification may be added. 

Q3. The performance of the membranes is only studied at a pressure of 1 bar while for Reverse Osmosis the pressure may go higher. So the studies at higher pressure may be reported.

Reviewer 3 Report

The authors present a manuscript with the title “Ionic Liquid-Mediated Interfacial Polymerization for Fabrication of Reverse Osmosis Membranes”, which gives novel insights into using ILs for the preparation of TFC membranes. The presented data and the drawn conclusions are consistent and well presented. This reviewer enjoyed reading this well prepared publication. The chapter concerning the water contact angle could be adjusted by taking the PA structure/flux data into consideration (see below). Besides this, this reviewer has no concern to recommend the manuscript for publication.

In chapter 4.3 the water contact angles of the different membranes are discussed. The differences in hydrophilicity are correlated to the number of probably free carboxylic functions, which is not unlikely. However, the authors already discuss different structures of the PA layer to explain their flux data. A different PA structure might also result in a variation of the surface structure (=roughness), which will also influence the wettability and could as well explain the differences in contact angle. SEM or AFM images of the membrane surface would be useful to further clarify this point.

Minor issues:

Line 174: there might be a double space in that line

Line 228/234/235/236: the unit “L/m2-hr” should be written as “L m^-2 h^-1” or just “LMH”

Line 247: “chain-length“ should be „chain length”

Line 269 “facilitated“ should be „facilitate”

Author Response

We thank the reviewer for appreciating our study. The point-by-point response is as follows:

Q1. In chapter 4.3 the water contact angles of the different membranes are discussed. The differences in hydrophilicity are correlated to the number of probably free carboxylic functions, which is not unlikely. However, the authors already discuss different structures of the PA layer to explain their flux data. A different PA structure might also result in a variation of the surface structure (=roughness), which will also influence the wettability and could as well explain the differences in contact angle. SEM or AFM images of the membrane surface would be useful to further clarify this point. 

Response: We agree with the reviewer that the differences in hydrophilicity are correlated to the number of free carboxylic functions, which in turn affects the membrane performances (flux and rejection efficiency). The trend was also confirmed using GIWAXS and zeta potential measurements. However, our AFM or SEM imaging of the different membranes fabricated in the study did not reveal any perceptible or concrete differences between the materials surface roughness that will affect the membrane performance.

We have not made any change in the manuscript in this respect.      

Minor issues:

Line 174: there might be a double space in that line

Corrected.

Line 228/234/235/236: the unit “L/m2-hr” should be written as “L m^-2 h^-1” or just “LMH”

Corrected.

Line 247: “chain-length“ should be „chain length”

Corrected.

Line 269 “facilitated“ should be „facilitate”

Corrected.